# Transition to Employment Program (SUPER) for Youth at Risk: A Conceptual and Practical Model

**DOI:** 10.3390/ijerph17113904

**Published:** 2020-05-31

**Authors:** Yonat Ivzori, Dalia Sachs, Shunit Reiter, Naomi Schreuer

**Affiliations:** 1The Israeli Ministry of Education, Jerusalem 91911, Israel ; yonativzo@gmail.com; 2Department of Occupational Therapy, Faculty of Welfare and Health Sciences, University of Haifa, Haifa 3498838, Israel; dalia.sachs@gmail.com; 3Department of Special Education, Faculty of Education, University of Haifa, Haifa 3498838, Israel; shunitr@edu.haifa.ac.il

**Keywords:** adolescents, knowledge translation, multidisciplinary intervention, model of human occupation (MOHO), meaningful learning, self-advocacy, future orientation, work skills

## Abstract

This article describes the development, implementation, and evaluation of the transdisciplinary “Successful Pathways to Employment for youth at Risk” (SUPER) program to raise the transition readiness of youth at risk (YAR) from school into participation in adults’ responsibilities and employment. More than 10% of adolescents are at risk of dropping out of school following poor academic attainments. Schools appraise academic merit but do not develop relevant educational programs to prepare youth to transition into adult working life. The SUPER program addresses a range of knowledge and skills required for this transition. Sixty YAR from three high schools participated in the 18-week SUPER program. Comparing the pre- and postintervention results revealed that the students’ engagement with responsibilities, objective knowledge about the work world, and self-rated self-advocacy skills improved as did their supervisor-rated work performance capacity. This study confirms the contribution of the SUPER model. Its concepts, tools, principles, and community labor-market involvement through workplace internships were effective and could apply in other contexts. The SUPER model provides evidence-based knowledge translation that can bring conceptual and practical changes towards successful transition and participation of YAR in adult working roles.

## 1. Introduction

Participation in employment in the open labor market is a major success measure for youth transition into adulthood. The transition process from school into adulthood is one of the most significant, complex, and multifaceted life-cycle processes [1,2]. It involves setting goals and making numerous decisions that may enhance career options in line with the individual’s future orientation. The transition into adulthood is especially complex for youth who experienced social or academic challenges or failure during their educational lives due to learning or other disabilities, family conditions, or socioenvironmental deprivations [3]. Youth at Risk (YAR) experience similar challenges that, in most cases, derive from learning disabilities, attention deficit, and emotional-behavioral disabilities. As a result, YAR have difficulty envisioning their futures as productive and successful adults [4,5,6]. Furthermore, YAR often lack family, social support, and role models and have little understanding of the process of transition to the world of work [7,8]. Additionally, in most countries, the focus of high schools is on academic achievement to enable graduates to access higher education [4,5,6]. Frequently, high schools do not have transition-to-work programs or special programs for YAR in place [9,10]. Consequently, these students become transparent to the school system and easily drop out of school. Eventually, they are pushed to the margins of society with long-term consequences in their adult lives. This population demonstrates further risk in not having future aspirations or plans. As one participant expressed, “Nobody has ever asked me these questions about my future aspirations; my concern is how to survive day by day” (M., 11th grade). The described challenges that YAR face call for developing transitional programs; however, most are based on a uni-disciplinary educational approach and focus mostly on the school systems’ environment. In a systematic review, Knight, Havard, Shakeshaft, Maple, Snijder, and Shakeshaft found that only 10% of 129 evaluation studies of interventions for high-risk young people addressed cooccurring risk factors, including disabilities, and they rated more than half of these studies as methodologically weak [11].

This paper presents a transdisciplinary program designed to facilitate the transition to employment of YAR and examines its effectiveness. The article describes the development and implementation of a transdisciplinary “Successful Pathways to Employment for youth at Risk” (SUPER) program that is based on the conceptual models of human occupations, education, and psychology. We will present the program’s principles and the research evaluating the evidence of the program’s contribution to the transition readiness of YAR into participating in continuing education and employment. 

### 1.1. Youth at Risk

In this research, we focus on YAR who study in special education classes integrated in regular educational framework and who exhibit unmotivated or disengaged behavior, truancy, inappropriate classroom behavior, and lack of basic vocational qualifications [12]. Students who are regularly absent from school report having low academic achievements, low quality of life [13], social-anxiety problems [14], and in some cases involvement in criminal behavior. Fifty percent of those cases result in full detachment, with the youth neither in school nor employed, and consequently, they are at risk of economic, social, and cultural detachment [15].

Scholars debate the term ‘youth at risk’ and its origins and definitions in various contexts [16,17]. In the present research, we use a definition based on the professional literature and the United Nations Convention on the Rights of the Child and that policymakers worldwide accept. That definition considers children and youth to be at risk if they live in family and environmental situations that endanger their ability to exercise their rights [6,18,19]. In the local context, the Israeli National Insurance Institute’s definition of youth at risk relies on seven identifiers: physical health and development, emotional health and well-being, social connectedness and participation, level of family support, education and skill acquisition, protection from others, and avoidance of risky behaviors [6]. Based on this definition, 16% of children and youth aged 0 to 17 years experienced risk situations [20]. The Organization for Economic Cooperation and Development (OECD) and the United States reported fairly similar data regarding personal and environmental risk factors affecting well-being among youth [5,21,22]. Australian researchers identified the most prevalent domains that occur among YAR as school absence, low levels of physical activity, unemployment, suicide ideation, mental distress, substance use, low health service utilization, and involvement in crime or with the juvenile justice system. In their research, they found all but one participant had experienced at least two cooccurring domains; more than half (58%) experienced four cooccurring domains. They further noted that the etiology of cooccurring risk factors is complex, being associated with a range of poor health, socioenvironmental determinants, childhood abuse, low socioeconomic status, and minority cultural identity [23]. 

### 1.2. Career Education Programs to Ease YAR Transition to Work

The relatively high prevalence of YAR in the population, their challenging personal and social backgrounds, and their difficulties integrating into the educational system and later into the labor market reveal the importance of developing support programs to help them successfully transition into adult life [9,10]. Ben Simon and Kahan-Strawczynski summarized three main barriers to integrating YAR into the workforce [4]. The first environmental barrier relates to the labor market, including rapid technological and organizational changes, as well as unstable employment. The second relates to significant differences between educational institutions and workplaces that foster misunderstandings in YAR regarding the process of transition to work [8]. The third barrier relates to the personal characteristics of YAR, such as low educational attainment dropping out, health situations affecting their abilities, immigration, and inadequate family support [5,24,25].

Other research suggested that youth, in general, and YAR, in particular, lack work skills, habits, and knowledge about labor rights, realistic job opportunities, and ways to pursue career development after school [26]. Some studies emphasized that YAR either lack future aspirations (related to idealistic ideation) or experience a large gap between their future aspirations and expectations (a more realistic ideation) [27]. Future expectations often decrease as youth gain greater understanding of their strengths and available opportunities, which affects their self-identity [7]. These barriers suggest the need for programs to facilitate the transition and adjustment of YAR to employment and adult life. However, little is known about effective, comprehensive ways to intervene with high-risk young people. Researchers have suggested that transition career programs for YAR should include knowledge about employment, job-search skills, vocational skills, self-awareness, self-advocacy, and normative behavior at work [28,29,30,31]. In addition, employers and youth indicated five key focus areas for transition programs: social skills (e.g., respect, conflict resolution); communication skills (e.g., verbal and nonverbal communication); high-order thinking skills (e.g., the ability to identify a problem, to obtain and evaluate information, and to problem solve); self-control (e.g., managing feelings and behavior); and positive self-perception and self-advocacy (e.g., presenting needs and strengths) [4,26]. All the reviewed programs addressed some or most of these content areas but did so from a mono-disciplinary (mostly educational) standpoint. However, the transition from school to work presents a more complex challenge that requires a transdisciplinary body of theoretical and practical viewpoints to establish translational knowledge for developing the contents (“what”) and methods (“how”) of effective transition program for YAR. 

Hence, we developed and evaluated the “SUPER program”, which aims to prepare and assist YAR in setting career-development goals, successfully transitioning from school to work and eventually participating in the labor market as productive adults. 

### 1.3. SUPER: Conceptual Framework, Program Principles, and Practices 

In the last decades, academic institutions and funding agencies increasingly have invested in and prioritized projects that integrate concepts, theories, methods, and approaches from multiple disciplines to address real-world problems innovatively and effectively [32,33]. This approach also yielded broader and more rapid knowledge translation [33]. Projects can be conceptualized along a continuum of increasing disciplinary integration and collaboration from uni- via multi- to transdisciplinary [34]. The SUPER program’s development and assessment reflects the transdisciplinary approach in which a team of researchers from different disciplines “aims to foster meaningful knowledge co-production through integrative and participatory processes that bring together diverse actors, disciplines, and knowledge bases” [35] (p. 30).

Three frameworks, each from a different discipline, underpin the SUPER program. The Model of Human Occupation (MOHO) is a leading frame of reference for theoretical concepts and evidence-based practical tools to understand and improve human participation in occupation [36]. It also has been applied to the work performance of at-risk populations. In analyzing occupational limitations, the MOHO model relates to four concepts. The term ‘volition’ includes a person’s beliefs, motivation, preferences, and desires that shape his or her (present and future) choices concerning doing and engaging in activities and occupations. The term ‘Habituation’ is the process by which occupation is organized into roles, patterns, and routines and enables the performance of student/worker roles. ‘Performance capacity’ relates to the physical and mental abilities and skills that underlie adequate enactment of activities. Finally, ‘environmental context’ refers to the physical and social environments in which occupation takes place. Additionally, the MOHO describes and explains how occupational adaptation processes occur and affect participation through changing life circumstances, such as the transition to employment.

Meaningful learning is a humanistic umbrella concept for numerous theories and models of teaching and learning [37], including the Cycle of Internalized Learning (CIL) [37,38]. The CIL, which we used in the current research, applies group interactions to problem solving. Group dynamics enable participants to develop interpersonal competences, such as cooperating, listening to different points of view, creating a nonjudgmental environment for the expression of opinions, and reaching group decisions democratically. In addition, CIL-based groups make their own rules and expect adherence to norms, such as adhering to time schedules, caring for the meeting place, and showing responsibility by following the group’s goals. The CIL approach suggests measuring outcomes of the group dynamics process not only in terms of functional achievements but also in terms of personality development. Important achievements include attaining a clear view of one’s self-image, personal interests, and aspirations. In addition, participants learn to respect personal boundaries; to develop self-identity; and to gain awareness and acceptance of others’ unique identities, needs, and opinions, as expressed in their ability to make decisions based on autonomy, self-determination, and self-advocacy. 

Rational Emotive Behavioral Therapy (REBT) is a psychological approach, a form of cognitive behavioral therapy that encourages development of rational thinking in order to facilitate healthy emotional expression and behavior. It is based on an understanding that how people think and perceive situations, which can be influenced by rational reflection, largely determines how they feel [39]. Ellis established three guiding principles for REBT—namely, adversity, beliefs, and consequences (ABC) [40]. *Adversity* relates to the activating nature of a challenging situation. *Beliefs* represents how the individual interprets the situation. *Consequences* refers to how the individual feels or acts, which depends on the nature of the adverse situation and the individual’s underlying beliefs. In the SUPER program, we used the ABC model with YAR to reinforce the feedback process for problem-solving and assimilation of work habits and skills.

We developed SUPER over two years and undertook preliminary research to evaluate it. We then implemented an updated SUPER in three schools over a second two-year period, 2016 to 2018. Multidisciplinary school teams, comprised of the classroom teacher together with the school’s educational advisor and occupational therapist, implemented SUPER in each school as a series of 13 weekly classroom meetings during one semester. Two of these meetings involved a field trip to workplaces. Following personal interviews regarding the students’ interests, wishes, and where they lived, the research and school teams found jobs for the students. Thus, the classroom component was followed by five working days (once a week over five weeks), during which participating students worked in the open labor market in those jobs. 

The program used various teaching methods: theoretical and experiential learning, simulations, problem-solving, meetings with managers, talks by employees and students who had successfully integrated into higher education and employment, visits to various industries, job analyses and observations, and work experience in the labor market. Based on the transdisciplinary conceptual framework, the SUPER program focused on four main content dimensions: Knowledge and understanding of concepts related to the working world and employability. The YAR participants acquired knowledge through lessons, meetings with employees, workplace visits, and especially through ongoing work experiences.Self- and occupational identity, which we conceptualized as participants developing a clear perspective of their current abilities, strengths, and desired future identities and learning skills related to self-advocacy and self-determination necessary to strengthen and act out such identities.Future orientation, which refers to the participants developing aspirations and addressing fears regarding adult lives through exposure to career planning, including higher education, professional education, and graduate studies.Work experience, which includes students’ participation in school and out-of-school duties, their development of performance skills and adherence to behavioral norms at school, and their experience of paid work in the labor market accompanied by ongoing feedback.

The SUPER program addressed the first three first dimensions during the classroom component. The fourth dimension, related to work participation, incorporated two parts: participation in school and home duties and five days of participation in their preferred job, escorted by both the school and research teams. The experience of a “real” job provided students an opportunity to implement the more theoretical parts of the program. 

In parallel with the school team’s program implementation, the research team evaluated its effectiveness in terms of the contribution SUPER made to participating YAR’s transitions from school to work. The research used five outcome measures for the program dimensions: (a) participation in school’s responsibilities and duties, (b) knowledge about the world of work, (c) self-advocacy, (d) future orientation, and (e) work performance. The overarching research hypothesis was that students would improve with respect to all outcome measures after (post-) participating in the SUPER program, compared to the evaluation before (pre-) participation in the program, specifically the following:H1: The students’ will report a higher level of participation in their school’s duties after participating in SUPER (e.g., performing duties and maintenance work at home, school, and volunteer; helping teachers and peers; employment status; and history).H2: The students will get a higher score on a knowledge assessment evaluation of concepts related to the working world after participating in SUPER (e.g., analyzing abilities vs. job demands, safety, and workers’ rights). H3: The students’ work and future career orientation evaluation score will increase after participating in SUPER (e.g., expressing dreams and plans regarding their future).H4: The students’ ability to advocate for themselves evaluation score will be higher after participating in SUPER (e.g., higher self-advocacy to represent their strengths and needs for accommodations).H5: The students’ work performance capacity score (participation at work) will improve while participating in SUPER. (e.g., appearance, job performance, dealing with authority, and teamwork during the work experience).

## 2. Materials and Methods

The research used mixed methods (qualitative and quantitative) for data collection and analysis [41]. This article describes only the quantitative method and results. We assessed the program according to evidence-based practice principles [42] and the Kirkpatrick model for evaluating training programs [43]. Assessment was performed by comparing outcome measures pre- and post-program participation.

### 2.1. Population

Following approval of the Northern District educational office, we selected three schools in Israel that offered special education classes for YAR and the school principal and relevant staff members consented to including the SUPER program in the curriculum. We obtained individual consent from all YAR candidates for the SUPER program and parental consent for each YAR to participate in the study and answer questionnaires. The consent letter explained the project and stipulated that all class members could participate in the program, regardless of whether they chose to participate in the research; that is, participation in the research was clearly not mandatory.

The research population consisted of 63 YAR students in grades 10 to 12. Three (5%) students dropped out of school and withdrew from the program; thus, pre–post data were available for 60 students (Table 1). Nearly all (90%) students were in grade 11, with an average age of 17.3 years (*SD* = 0.57), and the majority were male (41; 68.3%). Most (91.7%) of the students and their parents (75%) were born in Israel. 

### 2.2. Assessment Tools

Five tools assessed the contribution of SUPER. Respondents completed these tools pre- and postintervention, except where otherwise stated: The Background and Responsibilities Questionnaire collected data regarding demographics, the student’s participation in duties (school, home, and volunteer), and prevocational experiences (e.g., employment status and history).The Knowledge About the Working World (KAWW) questionnaire, which we adapted from the Concepts Questionnaire [44], measured knowledge and understanding of employment-related issues. It was composed of 16 short descriptions of work situations. The questionnaire required participants to match the situations with a concept (e.g., security issues, personal relations, legal rights, or environmental hazards) taught during the course. For example, participants should match the situation, “The supervisor did not allow the employee to take sick leave, despite the certificate provided by the doctor,” to the concept of rights.We developed the Self-Advocacy (SA) Questionnaire for Students and Teachers for the current study for use with students aged 15 years and older. The SA gathered appraisal data regarding the participants’ self-advocacy skills as assessed both by the participants and by their teachers. The questionnaire contained 15 items scored on a scale of 0 (*does not know*), 1 (*does not agree at all*), 2 (*partially agrees*), 3 (*agrees*), and 4 (*totally agrees*). We calculated the total score as well as scores for the three SA subscales: knowledge (about me, my environment, and my rights), social self-advocacy, and goal setting. Cronbach’s reliability values were, for the total SA (15 items), α = 0.69 (students) and α = 0.87 (teachers); for knowledge subscale (8 items), α = 0.65 (students) and α = 0.85 (teachers); for social self-advocacy subscale (3 items), α = 0.48 (students) and α = 0.78 (teachers); and for goal setting subscale (4 items), α = 0.79 (students) and α = 0.66 (teachers).The Future Orientation (FO) questionnaire collected self-reports from participants concerning their motivation (i.e., expectance, internal control, and external control), cognition (i.e., cognitive representations and the individual’s future), and behavior (i.e., exploration and commitment) with respect to work, career, and higher education [45]. The current study focused on five items pertaining to future work and career. For example, “How often do you think about or plan your future career?” Scoring options were 1 (*never*), 2 (*rarely*), 3 (*sometimes*), 4 (*often*), and 5 (*daily*). Cronbach’s reliability values were, for behavior (5 items), α = 0.75; for motivation (10 items), α = 0.71; and for internal control (4 items), α = 0.76.The work performance skills (participation at work) were measured by the Performance Capacity Card, assessing generic working skills that employers likely expect in employees, including pairs of items related to the concepts of attendance, persistency, engagement in teamwork, authority acceptance, job performance, initiative, safety, and independence at work [44]. The student and the supervisor conducted independent, parallel assessments of the students’ performance capacity with respect to eight items that were scored from 1 to 10 (where 1 = *fair performance*, 5 = *partial success*, and 10 = *full success*) and then were presented as a total mean score. The card scored attendance separately as a percentage of the students’ expected working hours. The Performance Capacity Card exhibited very good test–retest reliability when examined by *t*-test (*r* = 0.84–0.98), as shown by earlier testing on 20 employees on two occasions separated by two weeks [44]. Students and supervisors completed this card on the first, third, and fifth days (T1, T2, and T3, respectively) of the work experience component of SUPER.

### 2.3. Procedure 

The Head Researcher of the Ministry of Education and the Ethics Committee of the institution’s Faculty of Health and Welfare Studies (number 251/13) approved the research. The research team provided the school teams with program materials, trained them in conducting the program, and supervised the teams weekly throughout the program. 

A research assistant conducted the preintervention evaluation at each school prior to the commencement of SUPER and the postintervention evaluation after students and teachers completed the SUPER (Figure 1). 

### 2.4. Data Analysis

We used SPSS 23 (IBM SPSS Statistics for Windows, Chicago, IL, USA) to process the data, with significance set at *p* < 0.05. Descriptive statistics described the participants’ characteristics, and chi-square measured pre–post differences in students’ working frequency as well as in engagement with responsibilities in the school, home, and volunteering contexts. Paired *t*-tests compared the pre–post total scores of the three main outcome measures, as assessed by the KAWW, SA, and FO questionnaires. Repeated measure analysis of variance (ANOVA) was used to examine differences in performance capacity three times during the working experience. 

## 3. Results

### 3.1. Pre–Post Differences in Participation in Duties (School, Home, Work, and Volunteer) 

Participation in duties was measured by the background demographic and duties questionnaire. Pre–post comparisons using chi-square analysis (Table 2) revealed significant differences in employment participation status outside the SUPER program (χ^2^_(1)_ = 5.65, *p* < 0.05). Specifically, from pre- to postintervention, student employment increased from 25 (43.9%) to 47 (82.5%) of the 57 students who reported preintervention that they worked outside the SUPER program. Among those 25 students who reported positive employment status preintervention, the number who working weekly approximately doubled from pre- to postintervention. Specifically, the number of students who worked weekly grew from eight (33.3%) to 15 (62.5%). Furthermore, the number of students who worked only during school holidays dropped by about two-thirds. That is, they decreased from 10 (41.7%) students’ preintervention to three (12.5%) postintervention, although these differences were not significant (pre–post work frequency, χ^2^_(4)_ = 3.66, ns). A pre–post comparison of student participation in duties, as examined by chi-square, revealed significant differences with respect to school and home duties (χ^2^_(1)_ = 7.45, *p* < 0.01). Among the 57 students who reported having duties, the number engaging in school and home duties increased from 16 (28.1%) to 23 (40.4%) students. Thus, most of H1 is confirmed. In addition, seven students out of 57 started volunteering only after the program.

### 3.2. Pre–Post Differences in Knowledge of the Work World and Future Orientation

A pre–post comparison of student knowledge about the world of work, as assessed by the KAWW questionnaire, showed significant increases from pre- to post-program, confirming H2. Their future orientation, as measured by the FO questionnaire, also strengthened significantly in the cognitive, behavioral, and internal control subscales but not with respect to motivation, confirming H3 (Table 3). 

### 3.3. Pre–Post Differences in Self-Advocacy

At the preintervention assessment, there were no significant differences in students’ self-advocacy abilities between the students’ self-assessments and their teachers’ assessments of the students using the SA questionnaire. However, pre–post comparisons yielded significant differences. Students perceived significant improvement over the course of the intervention in their total self-advocacy abilities as well as on three of four SA subscales. Significant differences were found also for teachers but in the opposite direction. That is, teacher-rated SA scores were lower postintervention compared with preintervention. Overall, the results partially confirm H4 as related to students’ perception of their SA (Table 4).

### 3.4. Pre–Post Differences in Work Performance

A post hoc repeated measures analysis for students’ self-evaluation of their work performance, as measured by the Performance Capacity Card, indicated that all students evaluated themselves with a score of 10 (the highest score) in all three measures. There was therefore no difference between students’ self-rated scores across the three assessment times—the first (T1), middle (T2), and last (T3) days of the five-day work experience period. However, analysis of Performance Capacity Card data collected from their supervisors revealed significant differences between the three assessment times (Figure 2). As assessed by the supervisors, the students’ work performance improved significantly from T1 to T2 to T3 as measured in four work-performance areas: security (*p* < 0.001), performance, initiative, and independence (*p* < 0.001). Overall, the results partially confirm H5, only as related to supervisors’ assessment of the students’ participation at work. 

## 4. Discussion

Ben Simon and Kahan-Strawczynski well described the role of education in preparing students for the complex transition to work: “Preparation for employment should begin already in schools. In educational frameworks students acquire their ‘human capital’” [4] (p. 11). There, students are equipped with essential skills for adult life and employment, such as life skills, social skills, communication skills, and soft skills. An educational framework also can provide students with an opportunity for practical learning, whether through a dual model of theoretical learning alongside professional training or through an initiation [4,5].

High schools tend to emphasize academic achievements and frequently do not have transition-to-work programs in place, and there are no special community-based programs that could provide YAR specifically with comprehensive career education or preparation for career development [4,6,8]. Most researchers in the field have highlighted the significance of intervention programs for YAR and the implications for their academic, social, and mental well-being during their school years and future adulthood. Similarly, Knight et al.’s systematic review revealed that evaluation studies of intervention programs overlooked risk factors and were methodologically weak [11]. Many researchers have drawn attention to the lack of sufficient programs and evaluation research. Moreover, most existing transition programs are based solely on education models. They do not consider the complexity of the youths’ life stage or that this transition process is a phenomenon that exists at the intersection of education, the labor market, and transition to adulthood. As a result, most transition programs lack the ability to identify which of the various components that they include are essential [25]. The SUPER program was developed to address these challenges by establishing an evidence-based transdisciplinary model for transition programs. The findings and feedback from all stakeholders show how the SUPER program’s components and underpinning principles affect YAR. Overall, this study’s findings reveal that the SUPER program contributes to most outcomes of students’ transition to employment and participation at work, which can aid development of similar programs in other contexts. It is important to note that the findings should be cautiously interpreted. Being a field study in the educational system, the design was quasi-experimental. In addition, the pretest–postest design does not rule out possible treats to internal validity. 

Knowledge acquisition regarding the world of work in transition programs is the subject of debate in the literature and practice. Some researchers claimed that basic knowledge and values related to transition to employment should be explored in elementary and middle schools [46]. Others stressed that investigating and expanding students’ knowledge related to the labor market is an important part of transition programs [30,47]. They recommended conducting practical tours, observations, and meetings with workers, together with theoretical learning. Recent studies recommended adding broader components to programs, such as options for postsecondary education, for developing intimate relations, and for balancing family life and work (e.g., Reference [48]). 

This current study shows that participants’ knowledge improved during the SUPER program by implementing learning of work concepts with CIL active learning. Active learning through rich experiences made the work world more real to participants and raised their awareness of the relevance of employability to their near future (e.g., by teaching them to analyze their abilities and skills in relation to the job demands and risks). It provided the participants a sense of control (e.g., teaching about their rights and exploring opportunities) and improved their future orientation (e.g., helping them discover their options and exposing them to success stories). 

Self-advocacy and the development of self-advocacy skills are a major focus of the program, such that considerable time was dedicated to developing the participants’ ability to self-advocate, and various teaching tools were consolidated to that end. This approach is congruent with the literature concerning the essential components in successful programs and aligns with Lippman et al.’s and Ben Simon and Kahan-Strawczynski’s recommendations, in particular [4,26]. We measured self-advocacy from two viewpoints: those of program participants and of their teachers. The research revealed that program participants evaluated their self-advocacy abilities significantly higher at the end of the program than at the beginning. However, contrary to our expectations, their teachers evaluated students’ self-advocacy abilities as significantly lower at the end of the program than at the beginning on most measures (i.e., except for the goal-setting component). Qualitative findings, including research team observations, support the students’ self-assessments for self-advocacy (these findings will be presented in full in another article). Consequently, we conclude that students’ self-advocacy skills indeed improved over the course of SUPER, confirming H4. The unreliability of the teachers’ ratings in this specific instance may be rationalized on the grounds that the teachers had no knowledge base concerning student self-advocacy norms because self-advocacy is not part of the explicit or even hidden school curricula (unlike, for example, language skills and behavioral boundaries, respectively). Implementing the SUPER program allowed teachers an opportunity to get to know their students in depth and from unfamiliar angles and sharpened their awareness of some components of the students’ personalities of which they had been less aware at the beginning of the program. To overcome this in the future, researchers should utilize external subjective raters to assess student skills. 

The findings also indicate that participants increased the frequency of their engagement and showed improvement in the three components of future orientation: cognitive, behavioral, and internal control [45]. They acquired more knowledge concerning career options, developed hopes, expressed fears, and emotions regarding the possibility of succeeding in work and adult life and developed a sense of internal control rather than attributing their future to external factors, such as luck, economic conditions, social pressure, or other people.

Participation at work, defined also in terms of working skills and habits, is the main outcome in most transition-to-employment programs. Interestingly, from the students’ evaluations, this measure did not improve following participation in the program. Research shows that youth tend to evaluate themselves very high, regardless of their life experiences or achievements [49]. Similar to participants in other studies (e.g., Reference [50]), participants in the SUPER program evaluated themselves very high (10 out of 10) from the first time point on all work performance measures; hence, no statistical variance was found to show any change. However, the supervisors reported significant differences between the first, third, and final (fifth) days of work experience, with significant improvement in four work-performance items. In addition, although not significantly, the other four work performance items improved as well. This tendency might become stronger in a longer work experience. These performance skills that include behavior and performance at work as well as initiative and work relationships with peers and supervisors represent components that assemble into participation at work, as evaluated by the supervisors and students.

We believe that the method chosen for evaluating skills and habits by the two evaluator types (students and supervisors) was compatible: Supervisor ratings are necessary to obtain objective data with no variance when students themselves tend to over evaluate their performance. Nevertheless, students’ ratings are desirable and valuable feedback to assist the students in raising their self-awareness and in developing their executive functioning and self-monitoring skills. 

The SUPER program’s uniqueness lies in its development from a transdisciplinary theoretical and practical model suitable for assisting YAR students transition from school to the labor market and from adolescence to adulthood. It is considered a transdisciplinary model because it represents an approach whereby scholars and practitioners from various academic disciplines and professions worked collaboratively from the outset to synthesize and extend discipline-specific perspectives, theories, and methods. It is complex because it includes the various dimensions involved. The students are at its center; teachers and the school system come after. The workplaces and the community are the larger context in which the program takes place. 

We claim that the transdisciplinary model increases the prospects of program establishment and knowledge translation into an effective intervention program and thus addresses a complex social challenge. The underpinnings of SUPER were transdisciplinary, in that we developed it by integrating theoretic models from education, psychology, and occupational science and therapy (Figure 3). Its implementation was (a) multidisciplinary, in that school teams comprising educational, psychological, and occupational personnel implemented the SUPER; (b) multicontextual, in that SUPER was integrated into the school curriculum and implemented at high-school campuses and community labor markets (through visits and workplace internships); and (c) multireferent, in that we collected feedback from student participants, their teachers, and their supervisors. Finally, the SUPER had a multidomain scope, covering not only direct employment-oriented areas and participation at work (e.g., knowledge, skills, and work performance) but also areas related to personal development toward adulthood (e.g., occupational identity, future orientation, and self-advocacy).

Figure 3 presents a model of “how” to conduct the SUPER program and what methods to apply effectively among YAR. It visualizes the transdisciplinary approach on which SUPER is based and presents the learning dimensions covered. The SUPER program explored the first three learning dimensions gradually over several months to prepare the students for the fourth dimension, the participation in five working days in their preferred job. The school and research teams escorted the YAR during this transition. Participation at work greatly improved the YAR’s work skills and habits, as assessed by their supervisors. Indeed, the findings revealed that the SUPER program contributed to most outcome measures for participants’ transition to employment, supporting the integration of scientific and professional knowledge. After participating in the program, the YAR exhibited improved knowledge of work concepts, future orientation, and self-advocacy abilities; they engaged more in school duties; and their work skills improved, as perceived by their supervisors.

Using the MOHO model, which defines occupational performance and describes the essential component on which the program should focus, ensured that the SUPER related not to only one component, such as skills, volition (motivation), or habituation (development of roles and life procedures) but rather to all components in the context of the various environments in which occupations took place (the schools and community settings’ physical environments as well as the classroom and workplace social environments). Incorporating REBT cognitive therapy, CIL meaningful learning, and feedback cycles during simulations together with engagement in activities during real work experience led to improved outcome measures for participating YAR. 

## 5. Conclusions

The transdisciplinary approach enabled the development of an intervention program that enhanced the YAR’s skills, knowledge, and self-confidence in their transition to work and adulthood. The accompanying research and its findings provide evidence for the efficacy of the SUPER program in promoting successful pathways to employment. The project and research strengthen the importance of integrating transition-to-employment and work participation programs for YAR. Based on an interdisciplinary conceptual framework and professional teams, the program might be relevant to other students with disabilities in high schools. Although the research design is not experimental and calls for careful conclusions, the strength of the research is that it was implemented and successfully accepted in the field in which it takes place. 

Following the experience of including the SUPER program in high schools, the program success relies to a great extent on school staff being willing to collaborate with external training teams and appreciating the importance of incorporating transition-to-work and work experiences programming alongside academic studies. The study strengths other researchers’ claims that developing employability requires the integration of theoretical and experimental learning in the context of a real workplace [4,8]. This study shows that such a program should include professional training of the educational staff and should support them with ongoing adequate supervision throughout its implementation. 

Program implementation should help students to connect the school world with practical work experience and equip them with learning strategies and skills in both arenas as well as with knowledge about their future possibilities in the labor market. Therefore, program implementation requires creating partnerships between the educational system and the world of work in the local community. Furthermore, we recommend that the program expose students to the success stories of YAR graduates to increase motivation and to open opportunities for the future. 

The transition process from school to adulthood is one of the most significant, complex, and multifaceted life-cycle processes, especially for YAR. Therefore, it is important for school principals and educational and welfare policymakers to adopt the SUPER program with all its dimensions and stages to safeguard equal opportunity for YAR transition into productive members of society with aspirations and achievements.

This research was completed a short time after completion of the program. Limitations of the study design call for future replications of the program. In addition, future research should consider longer-term follow up of its graduates and should explore the suitability of the program for students from various disabilities and cultural and socioeconomic backgrounds. We currently are examining the program’s effectiveness among YAR in other societies and contexts, such as minority groups and in schools in Chicago, IL, USA.

## Figures and Tables

**Figure 1 ijerph-17-03904-f001:**
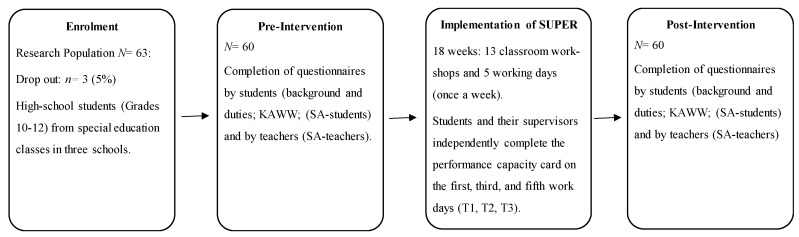
Research procedure flow chart. KAWW—Knowledge About the Working World; SA—Self-Advocacy.

**Figure 2 ijerph-17-03904-f002:**
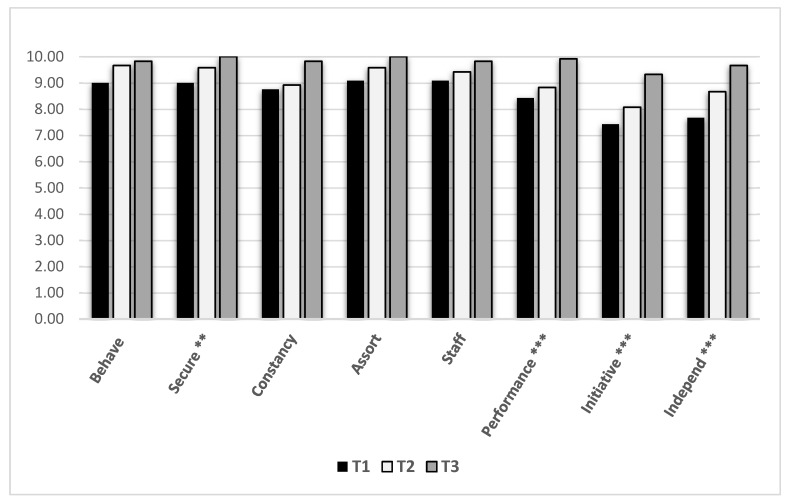
Supervisors’ perceptions of students’ work performance over time as measured by the Performance Capacity Card. ** *p* < 0.01; *** *p* < 0.001.

**Figure 3 ijerph-17-03904-f003:**
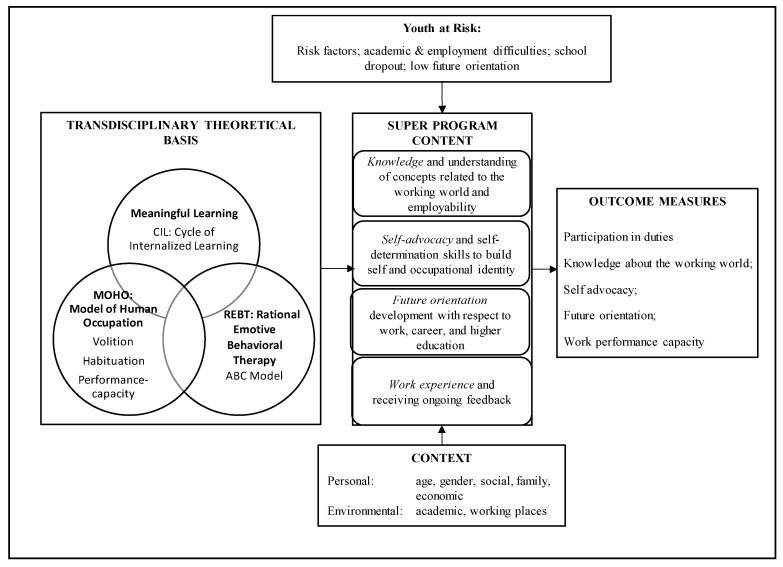
Model: Field approach to implementing the “Successful Pathways to Employment for youth at Risk” (SUPER) transition-to-work program.

**Table 1 ijerph-17-03904-t001:** Students’ (*N* = 60) characteristics and distribution across the three participating schools.

Characteristic	Category	*n*	(%)
The Schools	“M”	37	61.66
	“E”	16	26.67
	“T”	7	11.67
	Total	60	100
Grade	10	2	3.3
	11	54	90
	12	4	6.7
	Total	60	100
Gender	Male	41	68.3
	Female	19	31.7
	Total	60	100

**Table 2 ijerph-17-03904-t002:** Pre–post differences in extent of employment and engagement in fulfilling responsibilities.

Pre- or Post- Assessment	Working (*N* = 57)		Work Frequency (*N* = 24)						Engagement in Responsibilities (*N* = 57)	
	Yes		Weekly		Monthly		Holidays only		Yes	
	*n*	%	*n*	%	*n*	%	*n*	%	*n*	%
Pre-intervention	25	43.9	8	33.3	6	25.0	10	41.7	16	28.1
Post-intervention	47	82.5	15	62.5	6	25.0	3	12.5	23	40.4
χ^2^	0.032 *		0.454						0.02*	

* *p* < 0.05.

**Table 3 ijerph-17-03904-t003:** Pre–post differences in students’ future orientation (FO) and knowledge about the working world (KAWW), *N* = 55.

Outcome Measures	Pre-Intervention (T1)		Post-Intervention (T2)		*t* (*df*)	*p*
	*M*	*SD*	*M*	*SD*		
Future orientation: cognitive	3.38	1.250	3.870	0.900	−2.996 (54)	0.004
Future orientation: behavioral	1.96	0.731	3.060	0.674	−10.483 (56)	0.000
Future orientation: motivational	3.93	0.481	4.050	0.496	−1.628 (56)	0.109
Future orientation: internal control	4.14	0.595	4.400	0.539	−3.212 (56)	0.002
Knowledge about the working world	10.94	2.695	14.298	1.592	−8.915 (56)	0.000

**Table 4 ijerph-17-03904-t004:** Pre–post differences in the self-advocacy (SA) outcome measure as rated by students and teachers (*N* = 57).

Outcome Measure	Preintervention		Postintervention		*t* (*df*)	*p*
	*M*	*SD*	*M*	*SD*		
Student SA scales						
Total	2.93	0.397	3.08	0.341	−3.031 (56)	0.004
Knowledge	3.06	0.470	3.18	0.402	−2.090 (56)	0.041
Social	2.44	0.670	2.69	0.598	−2.995 (56)	0.004
Goals	3.01	0.766	3.16	0.710	−1.558 (56)	0.125
Teacher SA scales						
Total	2.84	0.566	2.41	0.941	3.764 (55)	0.000
Knowledge	2.82	0.641	2.38	0.990	3.545 (55)	0.001
Social	2.94	0.878	2.39	0.943	3.614 (54)	0.001
Goals	2.79	0.711	2.48	0.102	2.416 (55)	0.019

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
