# Peer review of "Transition to Employment Program (SUPER) for Youth at Risk: A Conceptual and Practical Model"

_ijerph, 2020, doi:10.3390/ijerph17113904_

Round 1

Reviewer 1 Report

Brief Summary

Using information from youth at-risk studies and challenged families, the authors of the article argues that the children are at a higher risk of various hardships in future employment and life success. Their family profile, and the deficiencies for educational attainment, will adversely affect the growth of these youths in many life areas.  The exposure to multiple disadvantages leads to low expectations, lack of self-advocacy, and overall hopeless perspectives.

The Model

The Successful Pathways to Employment for Youth at Risk (SUPER Model) is well defined and presented as founded on sound theoretical research and evidence-based models of a multidisciplinary nature. This inclusion of multiple fields highly strengthens the SUPER model and its endeavors. As an overview, the model program connects high school age at-risk students with a series of weekly classroom meeting and relevant field trips, involving a school team, a classroom teacher, an educational advisor, and occupational therapist. The classroom component is followed by five weeks of participating in a job. Pre-intervention and Post-intervention evaluations of youth and teacher perspectives provide potential changes to a view of future aspirations, future expectations, changes in self-advocacy, as well as other relevant feedback in both quantitative and qualitative measures.

Value of the Article

The theoretical value of the article is the attention it draws to various issues affecting the at-risk youths. The practical value of the article is it provides the foundation for an evidence-based model. This outstanding study provides for replication on an international level. The tables and visual models provide the reader with clarity in the study's intent.

The strengths and weaknesses are well fined and of particularly interesting note is in which teachers reported a lower level of youth self-advocacy at post-hoc. The authors note that such a response may be due to the measure of what the teacher felt the student rating was pre-test, or a reflection of the teacher’s own expectation for at-risk youth.

An Evaluation of the Reading

The entire article is written with the interests of the at-risk youth behavior in mind. The authors are clear in reference to the future potential of SUPER as in further post-hoc surveys after a set period. In addition, the study has future potential for control group comparisons as well as implementation for confined youth. The authors note that in addition to the quantitative results, qualitative measures not included in this particular article are available. Such data would be helpful for researchers wishing to replicate the study and/or build upon the model. Future studies could also include the comparison of risk factors to the SUPER MODEL program outcomes. In closing, this study has many strengths and the outcomes and intent is reminiscent of the high school vocational centers of the 1970s and 1980s that have since been terminated by many school systems for budgeting reasons. In many areas, the term “trade school” now refers to confinement locations for criminally active youth. The authors, in promoting this model, attempt to implement some of those evidence-based practices prior to negative events as an intervention rather than a reactive tool.

Author Response

Dear reviewer, 

We thank you for your support and encouraging review.

Naomi Schreuer (corresponding author) and the research team

Reviewer 2 Report

This paper presents a program to raise the transition readiness of youth at risk (YAR) from school into adulthood. The Authors applied the program to three high schools and evaluated it according to evidence-based principles. They say that they applied both qualitative and quantitative methods, but in the paper, they present only the latest.

The research question is clear and well defined, although it is presented in a very concise way. In any case, the paper is accurate, and it contains a good literature review concerning the topic. Especially the discussion of results is sound and well conducted, while the introduction is rather too long (half of the paper) and not clear.

I think that after a minor revision the paper will be interesting for the readership of the Journal, and it could attract a wide readership.

The work provides an advance towards the current knowledge, basing the conclusion on a case study that can be extended.

The paper does read fluidly, and it is written in American English.

  • Does the introduction provide sufficient background and include all relevant references?

According to me, the introduction is the weak part of the paper. It is too long (almost half of the paper) and it does not present, at the beginning, a clear idea of the topic of the paper, the method used and the results. I think that the paper will be really improved if the introduction is rewritten with more accuracy.                                               

  • Is the research design appropriate?

The research on which the paper is based seems well designed and it is based on a good literature review.                                            

  • Are the methods adequately described?

Methods are correctly described.

  • Are the results clearly presented?

Results are clearly presented

  • Are the conclusions supported by the results?

The conclusions are drawn appropriately based on the data presented.

Reviewer Comments to the Authors

Dear Authors, congratulations on submitting a welcome contribution. I really appreciated reading your paper, but I have some request for clarification:

- Abstract, line 19: I would skip descriptive statistic in an abstract.

- Introduction: As I have written above, I think that this section should be rewritten, giving much importance to the topic of the paper itself. For the general structure of the paper, I would extend a bit the description of the Hypothesis in this part.

- Paragraph 2 Materials and Methods, line 228 and following: the school where you applied the method are in the USA or Israel? Please specify.

- Figure 3 seem not well formatted.

Author Response

Dear reviewer,

Thank you for your support and your helpful comments that enabled us to improve our manuscript. 

In response to your feedback:

All corrections are colored in blue along the revised manuscript. 

We made minor corrections such as deleting statistics from the abstract and writing that the population is from Israel in the method section.

We re-organized and reduced the introduction to address the request “to put the ‘what the paper is about’ as the second paragraph in the paper, clear, concise and early”.

We redefined the hypotheses in measurable terms and added descriptive information.

We corrected figure 1 and 3 and grouped the items in the figures to make sure they fit the page layout.

In addition, we made few corrections in response to the special issue editors.

We used the concept of participation in the relevant context such as the program dimensions and the outcome measures. In addition, we elaborated on the students' description in relation to disability and special education.

Sincerely yours

Naomi Schreuer (corresponding author),  and the research team,

University of Haifa, Israel